# *Bacillus cereus* in Dairy Products and Production Plants

**DOI:** 10.3390/foods11172572

**Published:** 2022-08-25

**Authors:** Erica Tirloni, Simone Stella, Francesco Celandroni, Diletta Mazzantini, Cristian Bernardi, Emilia Ghelardi

**Affiliations:** 1Department of Veterinary Medicine and Animal Sciences, University of Milan, Via dell’Università 6, 26900 Lodi, Italy; 2Department of Translational Research and New Technologies in Medicine and Surgery, University of Pisa, Via San Zeno 37, 56127 Pisa, Italy; 3Research Center Nutraceuticals and Food for Health-Nutrafood, University of Pisa, 56127 Pisa, Italy

**Keywords:** dairy, *Bacillus cereus*, biofilm

## Abstract

Spore-forming *Bacillus cereus* is a common contaminant of dairy products. As the microorganism is widespread in the environment, it can contaminate milk at the time of milking, but it can also reach the dairy products in each phase of production, storage and ripening. Milk pasteurization treatment is not effective in reducing contamination and can instead act as an activator of spore germination, and a potential associated risk still exists with the consumption of some processed foods. Prevalences and concentrations of *B. cereus* in milk and dairy products are extremely variable worldwide: in pasteurized milk, prevalences from 2% to 65.3% were reported, with concentrations of up to 3 × 10^5^ cfu/g, whereas prevalences in cheeses ranged from 0 to 95%, with concentrations of up to 4.2 × 10^6^ cfu/g. *Bacillus cereus* is also well known to produce biofilms, a serious concern for the dairy industry, with up to 90% of spores that are resistant to cleaning and are easily transferred. As the contamination of raw materials is not completely avoidable, and the application of decontamination treatments is only possible for some ingredients and is limited by both commercial and regulatory reasons, it is clear that the correct application of hygienic procedures is extremely important in order to avoid and manage the circulation of *B. cereus* along the dairy supply chain. Future developments in interventions must consider the synergic application of different mild technologies to prevent biofilm formation and to remove or inactivate the microorganism on the equipment.

## 1. Bacillus Cereus Characteristics

### 1.1. Taxonomy and Growth Requirements

*Bacillus cereus* is a rod-shaped aerobic or facultative anaerobic Gram-positive, motile, spore-forming bacterium commonly found in the environment or as a contaminant of foods. *Bacillus cereus sensu stricto* (hereafter *B. cereus*) is the model organism of the *Bacillus cereus* group, also known as *B. cereus sensu lato*. This complex includes up to ten lineages of species that are genetically and phenotypically closely related [1,2,3] and comprises species with clinical, veterinary or agronomic importance, such as *Bacillus cereus*, *Bacillus anthracis*, the etiological agent of anthrax, and the most popular biopesticide *Bacillus thuringiensis*.

*Bacillus cereus* strains are able to grow in a moderately wide range of temperatures; most lineages can be considered mesophilic (showing an optimal growth temperature of 37 °C and survival below 10 °C), but psychotropic strains (with an optimal growth temperature below 10 °C and slow growth at 37 °C) have also been described, and therefore can be easily isolated from foods even under refrigeration. Some linages, including psychrotolerant, strains, are related to milk and dairy production environments [4].

*Bacillus cereus* vegetative cells can survive and replicate at pH values ranging from 5 to 10 and are quite resistant to salting, but are easily inactivated by thermal treating, such as pasteurization or cooking [5]. Differently, spores are highly resistant to various extreme conditions, such as high temperatures, freezing, drying and gamma- and UV-irradiation, allowing for the survival of *B. cereus* in the environment and on different surfaces. 

Biofilm production gives *B. cereus* an additional chance to persist and survive in hostile environments. Biofilms are bacterial communities firmly sticking to biotic or abiotic surfaces and are constituted by cells glued in a complex matrix of exopolysaccharides, proteins and DNA (extracellular polymeric substances, EPSs). The EPS protects microorganisms from a variety of environmental stresses (i.e., detergents and disinfectants; Oliveira Silva [6]) enabling sticking to surfaces, even in conditions where non-biofilm producers are commonly washed away by water or milk flux, and to resisting physical decontaminating treatments used in the industrial settings.

### 1.2. Bacillus Cereus Culture and Isolation

The microorganism, poorly demanded from a nutritional point of view, can be isolated on the most common culture media, but, in many substrates, it usually represents only a part of a mixed microbiota. For its selective isolation and count from food matrices, the selective media mannitol yolk poly-myxin B agar (MYPA) and polymyxin pyruvate egg yolk mannitol bromothymol blue agar (PEMBA) are often used. *Bacillus cereus* grows rapidly on these media (in approximately 24 h), generating colonies surrounded by an opaque halo due to the degradation of the lecithin present in the egg yolk by the lecithinases produced by the microorganism. 

These methods give a “presumptive” detection or count of *B. cereus*, as the methodology applied for routine analyses does not discriminate among the species belonging to the *B. cereus sensu lato* group. Further confirmation steps are then required for a precise identification (by biochemical or genomic tests), but this is not strongly demanded by the food business operators, aiming to monitor the diffusion of these bacteria to prevent products contamination.

Moreover, when evaluating *B. cereus* counts, the presence of vegetative cells or spores, or a mixture of both forms, should be considered. For the selective detection of spores, a thermal treatment (80 °C for 10 min) is applied [7,8,9]: this step exerts a double effect, namely the inactivation of vegetative cells and the activation of spore germination. When such additional analytical step is applied, care must be given to avoid the germination of spores in diluted samples before the application of the treatment; a short time sample processing is then required. 

If the samples are not submitted to thermal shocks, vegetative cells can start growing immediately on culture media, but spores can also germinate in a very short period, thus making it impossible to distinguish the two forms.

Many studies supply global *B. cereus* prevalence and count data without discerning between vegetative cells and spores; although a difference in pathogenicity between the two forms is evident (considering spores as quite metabolically inactive forms), the importance of the distinction depends on the substrate analyzed (more or less prone to *B. cereus* growth) and on the fate of the product (direct consumption or further transformation). The ratio between vegetative cells and spores in a food substrate strictly depends on the chemical–physical parameters, with a growing spore prevalence in harsh conditions (e.g., in ripened cheese). 

### 1.3. Disease Caused by B. cereus

*Bacillus cereus* was firstly isolated in 1887 and, for a long time, was considered as an environmental contaminant of poor importance, until it was recognized as pathogen for humans and other mammals. Nowadays, *B. cereus* is recognized to secrete a wide variety of virulence factors (toxins and enzymes), enabling the microorganism to be responsible for different infections, both localized or systemic. The genetic diversity of the *B. cereus* strains is reflected on the strains pathogenicity; some lineages, more adapted to cold temperatures and to dairy production environment, are not evidently linked to foodborne disease episodes [4]. Nevertheless, *B. cereus* is mostly famous for causing food-borne infections due to the ingestion of foods contaminated by bacteria, toxins or both, and is considered the etiological agent of the emetic or diarrheal syndromes. The diarrheal syndrome is caused by the ingestion of foods contaminated by *B. cereus* vegetative cells (Table 1). Once ingested, cells grow in the gut and actively secrete toxins. Haemolysin BL (HBL) and the non-haemolytic enterotoxin (NHE) are the major toxins involved in the diarrheal syndrome, although cytotoxin K (CytK) may also contribute [10,11]. HBL and NHE are trimeric toxins, whose components are independently secreted by bacterial cells and are able to associate in the extracellular environment. The complete toxins and the mono-component CytK act by inducing the formation of pores on cellular membranes [10]. The anaerobic or microaerophilic intestinal environment triggers toxin production, which causes microvilli destruction and cell lysis, inducing diarrhea [12].

The emetic syndrome is an intoxication due to the ingestion of a preformed toxin (cereulide) produced by *B. cereus* strains growing as contaminants in foods: mostly rice, pasta, milk or dairy products (Table 1). The enzymatic production of the dodecadepsipeptide cereulide requires the presence of non-ribosomal peptidyl synthetases encoded by a *B. cereus* megaplasmid [12]. Cereulide in foods is highly stable and resistant to thermal treating and proteolytic activity. After ingestion, cereulide induces nausea and vomit through vague nerve stimulation. The severity of intoxication depends on the amount of ingested cereulide and therefore on the total number of *B. cereus* cells in the food, which may vary from to 10^3^ to 10^5^ cfu/g [13]. Although all subjects are equally susceptible to this intoxication, the most severe symptoms have been described in the young and elderly [14,15,16].

### 1.4. Reservoir and Contamination Routes

*Bacillus cereus* is widespread in the environment. The main environmental reservoirs are represented by decaying organic matter, fresh and marine waters, plants and fomites, but the pathogen is also naturally present in the intestines of various animals, resulting in soil contamination. Due to its ubiquity, this microorganism can easily contaminate a very assorted group of foods and it is virtually impossible to obtain raw materials that are free from *B. cereus* spores [17,18]. Hong et al. reported that the concentration of spore-forming bacilli is approximately 6 log spores/g in soil and 4 log spores/g in the human feces, suggesting a possible adaptation as gut commensals [19].

*Bacillus**cereus* is often isolated from plants used as primary food (cereals, and, in particular, rice) as a consequence of soil contamination. The survival of the spores to heat treatments (e.g., cooking) determines the potential risk associated with the consumption of cooked dishes stored at inadequate temperatures, which allow for the germination, multiplication and production of cereulide. Contaminated vegetables can also be used as ingredients (vegetables, spices) in the food industry, causing the contamination of originally non-contaminated foods. However, the range of potentially contaminated foods is very wide. In fact, *B. cereus* has been isolated from dairy products, pasta, rice-based dishes, meats and derivatives, soups, pastry products and sauces. It can also be frequently isolated from the production environments of many types of foods, thus making it possible to contaminate them during all stages of production.

## 2. Role of *B. cereus* in the Dairy Supply Chain

### 2.1. Presence of B. cereus along the Dairy Production Chain

As a ubiquitous microorganism, *B. cereus* can be isolated in all of the steps of the dairy production chain. At the farm, the presence and diffusion of *B. cereus* is associated with the contamination of soil, fodder and bedding, which are in contact with the udder of the cows. At the time of milking, spores are conveyed in the milk and can survive the pasteurization treatment, which cannot be considered an effective hurdle, otherwise acting as a potential activator of their germination. In the presence of favorable environmental conditions, the vegetative forms are consequently able to replicate in the dairy substrate, likely determining the spoilage of the product or sometimes being a risk for consumers. The effect of the contamination on the product quality and safety is then determined by the characteristics of the dairy products considered (see Section 2.3).

Considering that spores are present in raw milk, it is expected that they will be present in the whole production process [20], with multi-sources of contamination that contemporarily coincide. As highlighted by Wirtanen and Mettler [21,22], the dairy production system is extremely complex and is characterized by dead ends, pockets and traps that are difficult to clean and that can harbor a permanent reservoir for spoilage bacteria (e.g., valves, shafts and gaskets). Milking equipment can also be a contamination source [23,24], with spores that may adhere to and germinate in dairy equipment (e.g., tanks, pasteurizers, packaging machines), causing the post-treatment contamination of milk [18,25,26]. The presence of spores on the surfaces in direct contact with the products and in the processing plants is favored by their structure. Their adherence to food contact surfaces (including stainless steel) is mainly due to a highly hydrophobic external covering, the exosporium, whereas a minor role is played by other mechanisms, such as proteic appendages and relative surface charge [27]. Suitable temperatures, a high humidity and the presence of a liquid or air interface are favoring factors for the germination of the microorganisms and for the consequent production of biofilms [28].

### 2.2. Spoilage and Pathogenic Potential of B. cereus

Members of the *B. cereus* group are among the main organisms responsible for the spoilage of dairy products; thus, they are carefully considered for the possible contamination of the production plants [29]. Food spoilage due to *B. cereus* is often observed in heat-treated products, where the microorganism survives through the spores. In the presence of heat activation in ideal environmental conditions, some dormant spores are able to germinate [30], whereas some others regress to their dormant form.

*Bacillus**cereus* strains are generally able to produce extracellular or intracellular thermo-resistant enzymes (proteases, lipases, phospholipases) that are involved in milk and dairy product spoilage (e.g., off-flavor, rancidity, bitterness) [31]. Prevalences and concentrations (cfu/g) of *B. cereus* in milk and dairy products are extremely variable worldwide (Table 2).

Among analyzed raw milk (from Egypt and Turkey), the reported prevalence was very variable, ranging from 3.8 to 100%, with concentrations of up to 10^5^ cfu/mL, whereas, in pasteurized milk (from Netherlands, Denmark, Taiwan, India, Egypt, Turkey, Canada, China, Poland and Japan), the prevalence ranged from 2 to 65.3%, with concentrations of up to 10^5^ cfu/mL (Table 2). These data confirm the ineffective action of milk pasteurization in reducing the prevalence and count of this microorganism, thus requiring the food business operator to carefully manage the milk during its shelf life.

Considering ice creams, the reported incidence (data obtained from products from Taiwan, Germany, Egypt and India) ranged from 25.0 to 62.8%, with concentrations of up to 10^8^ cfu/g. In this case, it has to be considered that the ice cream production process, though manufacturing operations usually include the pasteurization or heating of the mixture of ingredients, does not include a decontamination step, and the *B. cereus* counts found in the products can be considered comparable to those present in the milk used for the production. Among milk derivatives, particular attention is paid to powdered milk intended for infants. The milk dehydration process does not exert significant decontamination; the spores, which are dormant in the powdered product owing to low activity water values, can rapidly germinate when the milk is reconstituted at permissive temperatures (near 37 °C) and, in the case of prolonged storage, high levels of *B. cereus* can be reached, with a consequent risk for this category of particularly vulnerable consumers. Several studies showed that spores of *B. cereus* can be found in milk powder and infant formulas (Table 2), with prevalence from 6.8 to 68% and loads that reach up to 10^4^ cfu/g. The US Dairy Export Council has set strict limits for mesophilic (<3 log cfu/g) and thermophilic spores (<2.7 log cfu/g) in dairy powders, pointing to the importance in reducing the factors that lead to their proliferation and survival in processing plants [52]; European legislation (European Commission, EC, 2005, Reg. 2073/2005) considers presumptive *B. cereus* as a process hygiene criterion for dried infant formulae (EC, 2005) [53].

Cheeses showed a very high heterogeneity in terms of contamination, with reported prevalences from 0 to 95%, and concentrations of up to 10^6^ cfu/g; this high variability can be explained by the differences among cheese typologies: owing to the gradual decrease in water activity during ripening, it could be expected that, independently from the prevalence, high counts could be revealed only in fresh unripened cheeses.

Besides showing spoilage abilities, some *B. cereus* strains are emerging pathogens that carry innate resistance to antibiotics or produce toxins. Borge [54] showed that most strains isolated from dairy products are cytotoxic and possess *nhe* and *hbl* genes, but they were rarely able to produce cereulide. Svensson [55] found a very low prevalence of emetic strains (1.5%) out of 5668 *B. cereus* isolates from milk obtained from 10 different dairy products. Tirloni [56] showed that all eleven isolates from the dairy processing plant of PDO Taleggio cheese produced phosphatidylcholine-specific phospholipase C (PC-PLC), haemolysins and proteases, and that most of them produced HBL (66.7%); all of the clusters harbored the *nheA*, *sph*, *plcA*, *entFM* and *cytK* genes, and some also harbored *nheB* (83.3%), *nheC* (66.7%), *bcet* (50.0%) and *entS* (66.7%). Catania showed that all fourteen isolates from processed cheese were positive for *nheABC*, *entFM* and *cytK* genes, whereas *hblABCD*, *bceT* and *ces* were not detected [57]. In 2015, Hwang and Park showed that 14% of sixty-nine *B. cereus* isolates from milk powder in Korea carried *hbl*, nhe, *cytK*, *bceT* and *entFM* genes, and 30% showed strong haemolytic activity [58].

### 2.3. Growth Potential of B. cereus in Dairy Products

Dairy products are characterized by heterogeneous substrates that may affect the growth of *B. cereus*, whose presence results from the contamination of milk or as post-process contamination. In previous studies, different categories of milk and milk products were selected to evaluate the potential growth ability of *B. cereus* strains (Table 3). Pasteurized milk was demonstrated to be an optimal substrate for *B. cereus* growth due to the inactivation of microorganisms and enzymes [59,60,61]. Contrarily, in raw milk, no *B. cereus* growth was detected (<0.5 log cfu/g) due to the action of the present natural microflora (mainly composed by lactic acid bacteria) that, coupled with substrate acidification, exerted a competition [60]. A similar trend was observed in unflavored yogurt, where the very limited growth of inoculated *B. cereus* was observed (<1 log cfu/g from the day of inoculation until after 5 days of storage) due to a low pH and the presence of starter cultures [60]. Fresh ricotta, primo sale cheese and mascarpone were reported as extremely suitable substrates for the replication of *B. cereus* [60,62,63].

In Taleggio cheese and ricotta salata cheese [9,60], no *B. cereus* growth was detected (<0.5 log cfu/g) due to the action of the natural microflora (mainly composed of lactic acid bacteria) and due to the ripening that determined inadequate physical–chemical conditions (pH 5.8–6.8 in Taleggio cheese, pH 5.30–6.32 in ricotta salata cheese after 90 days of ripening). The same trend was observed by Rukure and Bester [65] during the ripening of Gouda cheese, which resulted in not offering proper conditions for *B. cereus* growth.

### 2.4. Biofilm Formation

Spore-forming bacteria such as *B. cereus* have been continuously detected throughout the dairy processing, including dairy farm environments, storage tanks, transportation tanks and dairy production plants [66,67]. *B. cereus* is well known to produce biofilms that offer a source of spores contamination for food during production stages [68]. According to Flemming and Wingender [69], biofilms, including those produced by *B. cereus*, are microbial communities usually embedded in a self-produced matrix whose structural elements are exopolysaccharides, proteins and DNA. *Bacillus*
*cereus* biofilm is mainly composed of vegetative cells, even if previous studies demonstrated that sporulation could occur in biofilms, confirming biofilms as an important source of spores’ contamination [70]. In the same study, the tested *B. cereus* strains were able to form mono-species biofilms on stainless steel, with up to 90% spores after 48 h of incubation. These spores resulted in being highly resistant to cleaning and, contemporarily, were easily transferred to agar, indicating a very fast aptitude to cross-contaminate food.

As for other biofilms, those produced by *B. cereus* may include several other bacterial species becoming a cooperative conglomerate in which each component may contribute to the community resilience and expansion [71,72]. This is favored by the complex matrix of exopolysaccharides, proteins and extracellular DNA, which are necessary for the adhesion on different materials. Pereira-Perez-Alonso [73] investigated the behavior of *Listeria monocytogenes* in mono-species biofilms co-cultured with *B. cereus*, highlighting a co-operative behavior between the two microorganisms, but showing that, in the presence of antagonistic substances produced by *B. cereus*, the counts of *L. monocytogenes* were lower if compared to mono-species biofilm counts.

Biofilm is a survival strategy for *B. cereus*, acting as a protection from harsh and stressful conditions, including physical and chemicals stresses (e.g., sanitizers) or antimicrobials [74,75]. According to Kwon [76], the food-industry-related conditions could promote *B. cereus* biofilm formation. Wijman [77] investigated the biofilm production of 56 strains, showing that it is strongly dependent on the incubation time, temperature and medium, as well as on the strain, with some strains able to produce biofilm in 24 h with subsequent dispersion within the next 24 h. In the same study, *B. cereus* showed thick biofilms developed at the air–liquid interface, whereas, in submerged systems, biofilm production was lower, suggesting that *B. cereus* biofilms may develop particularly in industrial storage and piping systems that are partly filled during operation, or where residual liquid remain after a production cycle. In general, depending on the strain, *B. cereus* can grow as immersed or floating biofilms and is also able to produce, within the biofilm, metabolites, surfactants, bacteriocins, enzymes and toxins that can have repercussions directly on the biofilm itself and/or on the environment [72]. In another study, Hayrapetyan [78] investigated biofilm formation by *B. cereus* reference strains ATCC 14579 and ATCC 10987 and 21 wild food isolates on polystyrene and stainless-steel tube as contact surfaces, highlighting an enhanced forming capacity when in contact with the stainless-steel tube compared to polystyrene.

The presence of spore-forming bacteria as biofilm formers in dairy plants is of great concern. Pereira-Perez-Alonso [79] explored the diversity of spore-forming bacilli in 100 samples of skimmed and whole UHT milk, finding the presence of *B. cereus* able to form biofilms in seven samples. Catania [57] examined processed cheeses in an Italian dairy plant, isolating *B. cereus*: five of the eight strains resulted in being strong biofilm producers. Fei [80] analyzed five hundred milk powders, isolating forty-two *B. cereus*: 66.7% of these showed a strong ability to form biofilm on the stainless steel tube. Radmehr [81] evaluated the characteristics of *B. cereus* in raw and pasteurized milk samples collected in Victoria (Australia), showing that most of the isolates (53.7%) possess a weak biofilm-forming capacity, with only 4.9% being strong biofilm producers. In the same study, a higher number of biofilm producers was detected among isolates from pasteurized milk compared to raw milk.

The detection and control of biofilms in the dairy industry are crucial in the spoilage control of heat-treated products. In the study by Oliveira Silva [6], the efficacy of peracetic acid and sodium hypochlorite against biofilms induced on stainless-steel surfaces was evaluated, showing on a pilot scale a reduction in its formation after treatment. Differently, Lin [82] found that two out of fifteen strains were capable of forming biofilm that was effectively inactivated by hot-acid or hot-alkali, while ten isolates showed tolerance to acid and alkali when associated in biofilms. As mentioned above, biofilm production is favored by the presence of equipment/structures that are difficult to reach during routine cleaning procedures (especially in closed systems), confirming that biofilm is a great challenge in the dairy industry. *Bacillus* biofilms were found to be detectable on both stainless-steel and Teflon surfaces after all cleaning procedures [21]. Even with adequate cleaning-in-place (CIP) systems that are frequently used in the dairy industry, some microorganisms could be found after cleaning (te Giffel et al., 1996; Wirtanen [18,21]. In fact, CIP systems showed a different efficacy in eliminating biofilms, with alkali-based CIP, which seems to improve the removal compared to CIP, usually used in the dairy industry [37]. These authors applied a response surface methodology, an effective method used to optimize process conditions, showing that the optimized CIP regime (1.5% NaOH at 65 °C for 30 min–water rinse–1% HNO_3_ at 65 °C for 10 min–water rinse) allowed the obtaining of a higher reduction in *B. cereus* biofilm cells compared to reference CIP (1% NaOH at 65 °C for 10 min–water rinse–1% HNO_3_ at 65 °C for 10 min–water rinse) (4.77 log/cm^2^ vs. 3.29 log reduction/cm^2^, respectively). Considering the potentiality of a natural food additive, Kang [83] investigated the effect of thyme essential oil on biofilm formation by *B. cereus*, showing a promising inhibitory effect.

## 3. Contamination Control Strategies Applicable to the Dairy Supply Chain

### 3.1. Reduction in Contamination in Raw Materials

The contamination of raw materials is not completely avoidable, but the collection and storage procedures can limit the concentration of contaminating bacteria. The application of decontamination treatments (e.g., irradiation, fumigation, heat treatment) is only possible for some ingredients (e.g., spices) and is limited by both commercial and regulatory reasons. The contamination of milk during milking can be severely limited with meticulous compliance with hygienic procedures; in order to avoid spoilage, it is recommended to maintain values of *B. cereus* below a threshold of 3 log cfu/mL.

### 3.2. Removal of Spores from Surfaces and Production Equipment

The sanitation procedures applied in the production plants must be focused on two main objectives, which are the removal of the biofilm and the inactivation of spores. As mentioned above, common cleaning and disinfection protocols, both on “open” equipment and through CIP, are not always sufficient to achieve these objectives. Some innovative approaches were therefore proposed: for a more effective removal of the biofilm, different enzymes (protease, polysaccharidases) were tested, which disaggregate the biofilm structure, allowing for its removal and favoring the subsequent action of disinfectants against the contained microorganisms. However, this approach is limited by the high costs of the necessary compounds. A second approach instead aims to “outflank” the natural resistance of spores to sanitizing treatments, in particular, by determining their germination (and then allowing the disinfection agents to act on the vegetative forms) [84,85].

### 3.3. Product Conditioning

The creation of unfavorable conditions for the germination of *B. cereus* spores and for the development of vegetative forms, through the application of mild technologies, can have a significant positive impact on food safety, preventing the possible presence of toxins. Food acidification, as well as salting/seasoning (by a reduction in water activity) represent effective hurdles. An interesting approach, especially in the dairy area, is the use of mixtures of lactic acid bacteria for bio-preservative purposes. A study [56] demonstrated the inhibiting effect of autochthonous lactic acid bacteria in fresh cheeses, with a reduction in the *B. cereus* count by approximately 100 times.

The possibility of decontaminating treatments should also be considered. This approach, however, is very tricky, since *B. cereus* spores are very resistant to heat (wet and dry) and to chemical, physical and radiation treatments. In a previous study [63], food grade organic acids were tested as antagonists for *B. cereus* growth. A concentration of acetic acid of 24.98 mM and a concentration of 22.20 mM of lactic acid were effective in broth at 30 °C, but, when applied to fresh cheese (on the surface of Primo Sale cheese), *B. cereus* counts did not show any difference at 15 °C among samples treated with organic acids compared to the control series, confirming the difficulty in exerting an effective action in complex matrices.

The possibility of applying heat treatments, such as sterilization, to dairy products is very limited, as it can determine evident sensory modifications, but the combination of different milder techniques can exert a synergistic action. Recent studies have demonstrated the possible efficacy of high hydrostatic pressures (HHP) in inactivating bacterial spores when combined with moderate temperature heat treatments (>60 °C) [86]. The sterilization of a final product can effectively reduce the risk associated with the presence of spores or vegetative forms, but it cannot prevent the risk associated with the presence of preformed emetic toxin.

### 3.4. Food Management

One of the simplest way to prevent *B. cereus* replication is to control temperatures in the preparation and storage of food. Different cooking methods allow for the destruction of vegetative cells and spores (steaming, roasting, frying or grilling), but cooking at low temperatures (below 100 °C) allows spore survival. One of the main causes of *B. cereus* toxinfection is the incorrect maintenance of cooked foods, which must be managed by avoiding the maintenance at permissive temperatures, e.g., ensuring constantly high temperatures (“hot bond”) or rapid cooling. In the case of cooling, after a heat treatment, it is indicated to reduce the temperature from 57 °C to 21 °C in less than two hours and to reach 5 °C in no more than six hours [87]. The production of enterotoxins and emetic toxins can take place starting from approximately 10 °C and 12 °C, respectively; therefore, although some strains are psychrotrophic, only inadequate refrigeration can result in a consistent production of toxins. The correct management of heat treatments and processing and storage temperatures represents a focal point of HACCP plans, but major problems occur when preparing meals at sales/administration establishments or in the household management. The best strategy, in this case, is therefore represented by the correct training of operators and consumer information.

## 4. Conclusions

The control of the presence of *B. cereus* in dairy production plants is crucial for limiting the replication of the microorganism. The application of good manufacturing practices (GMPs) is extremely important in this sense to minimize the risk associated with the presence of *B. cereus*. The traditional hygiene procedures applied in the food industry seem to have limits in the control of surface and equipment contamination with spores and biofilms. All future efforts should be addressed toward deepening the study on how *B. cereus* biofilms are built on their contribution to spoilage and pathogenicity and how to reduce their presence in the environment. As the prevalence of *B. cereus* on dairy products can be limited but not completely avoided, further efforts should also be addressed to prevent the germination of spores in the products. In this field, mild technologies, specifically designed for different dairy products, may be convenient to reduce the risk correlated to the presence of the pathogen. The same care should be taken for both fresh products (e.g., soft cheeses), which are prone to *B. cereus* growth, and for ripened cheeses (to prevent the early formation of stable cereulide). Examples of useful hurdles are temperature, acidification and bacterial competition, which should be combined in a strictly defined intervention applied through the production process.

## Figures and Tables

**Table 1 foods-11-02572-t001:** General traits of the diarrheal and emetic syndromes caused by *B. cereus*.

	Diarrheal Syndromes	Emetic Syndrome
**Type of toxin**	Proteins (HBL, NHE, CytK)	Cyclic peptide (cereulide)
**Site of production**	Small intestine	Preformed in food
**Incubation period**	8–16 h (up to 24 h)	0–5 h
**Disease duration**	12–24 h (sometimes > 24 h)	6–24 h
**Infectious dose**	10^5^–10^7^ ingested total cfu	10^5^–10^8^ cfu/g of contaminated food
**Resistance to heat**	Weak	Highly stable (up to 121 °C for 90 min)
**Symptoms**	Abdominal pain, watery diarrhea, nausea (sometimes)	Nausea, vomit, general weakness, diarrhea (sometimes)
**Involved foods**	Meat and derived foods, vegetables, sauces, soups	Rice, pasta, pastry products

**Table 2 foods-11-02572-t002:** Prevalence and concentrations of *B. cereus* in milk and milk products reported in previous studies worldwide.

Reference	Products *	Prevalence (Positive/Total Samples Analyzed)	Counts (cfu/g)	Country
[8]	raw milk (s)	60% (2/53)	6.3 × 10^2^–2.4 × 10^3^	Egypt
[32]	raw milk (v + s)	3.8% (11/106)	4 × 10^1^–3.8 × 10^5^	Turkey
[33]	raw milk (v + s)	100% (25/25)	10^1^–2.2 × 10^2^	Egypt
[34]	raw buffalo milk (v + s)	33.3% (50/150)	*Data not present*	China
[18]	pasteurized milk (v + s)	33% (56/157)	10^0^–10^4^	Netherlands
[35]	pasteurized milk (v + s)	56% (257/458)	10^1^–3 × 10^5^	Denmark
[36]	pasteurized milk (v + s)	2% (1/200)	2.8 × 10^2^	Taiwan
[37]	pasteurized milk (v + s)	55% (30/55)	10^1^–10^4^	India
[8]	pasteurized milk (s)	15% (3/20)	2.0 × 10^1^–2.0 × 10^2^	Egypt
[32]	pasteurized milk (v + s)	26% (13/50)	10^1^–1.1 × 10^3^	Turkey
[38]	pasteurized milk (v + s)	41% (104/254)	Up to 10^5^	Canada
[39]	pasteurized milk (v + s)	27% (70/258)	Mean: 11 MPN/g	China
[40]	pasteurized milk (v + s)	30% (18/60)	2 × 10^0^	Poland
[41]	pasteurized milk (v + s)	65.3% (66/101)	*Data not present*	Japan
[34]	pasteurized buffalo milk (v + s)	15.3% (46/300)	*Data not present*	China
[33]	uht milk (v + s)	13.3% (2/15)	10^2^	Egypt
[33]	condensed milk (v + s)	33.3% (3/10)	10^2^–4.0 × 10^2^	Egypt
[36]	fruit-flavored reconstituted milk (v + s)	2% (1/200)	1.5 × 10^1^	Taiwan
[36]	fermented milk (v + s)	17% (34/200)	5.0 × 10^0^–1.2 × 10^2^	Taiwan
[36]	ice-cream (v + s)	52% (104/200)	5 × 10^0^–2.5 × 10^2^	Taiwan
[36]	soft ice-cream (v + s)	35% (70/200)	5 × 10^0^–8.0 × 10^2^	Taiwan
[42]	ice-cream (v + s)	62.8% (508/809)	0.1–2 × 10^1^	Germany
[8]	ice-cream (s)	25% (10/40)	5.2 × 10^2^–1.5 × 10^3^	Egypt
[37]	ice-cream (v + s)	40% (10/25)	10^2^–10^8^	India
[36]	milk powder (v + s)	27% (54/200)	5 × 10^0^–4.5 × 10^2^	Taiwan
[7]	dried milk products (s)	46% (175/381)	3 × 10^0^–10^4^	Chile
[43]	infant formula (v + s)	18.3% (11/60)	<10^6^	Italy
[37]	milk powder (v + s)	52% (18/35)	10^2^–10^3^	India
[8]	full-fat milk powder (s)	15% (3/20)	10^1^–3.9 × 10^2^	Egypt
[8]	infant formula (s)	10% (2/20)	4.0 × 10^1^–2.1 × 10^2^	Egypt
[44]	infant formula (v + s)	6.8% (40/587)	10^3^–10^4^	China
[45]	infant formula (v + s)	7.53% (501/6656)	10^1^–5 × 10^3^	China
[33]	milk powder (v + s)	68% (17/25)	10^2^–3.5 × 10^3^	Egypt
[8]	yoghurt (s)	0% (0/20)	-	Egypt
[46]	Fermented milk (nunu) (v + s)	35.7% (10/28)	Mean: 6.5 × 10	Ghana
[37]	butter (v + s)	20% (5/25)	10^3^–10^4^	India
[47]	port salut argentino cheeses (v + s)	50% (15/30)	*Data not present*	Argentina
[47]	quartirolo cheeses (v + s)	0% (0/20)	-	Argentina
[37]	cheese *(paneer)* (v + s)	4% (1/25)	2 × 10^1^–4 × 10^1^	India
[37]	cheese *(khoa)* (v + s)	0% (0/20)	-	India
[37]	cheese (v + s)	33% (8/25)	10^2^–10^6^	India
[9]	ricotta salata cheese (s)	33.3% (48/144)	2 × 10^1^–2.6 × 10^2^	Italy
[46]	cheese (*west african soft*) (v + s)	38.7% (12/31)	Mean: 4 × 10^2^	Ghana
[32]	cheese (v + s)	10.4% (11/106)	4 × 10^1^–3.8 × 10^5^	Turkey
[40]	fresh acid cheeses (v + s)	8.6% (3/35)	10^1^–1.6 × 10^2^	Poland
[40]	mold cheeses (v + s)	52.5% (42/80)	10^2^–2.0 × 10^3^	Poland
[40]	ripening rennet cheeses (v + s)	43.4% (76/175)	10^1^–6.5 × 10^3^	Poland
[48]	soft stretched curd cheeses (v + s)	24.5% (81/331)	<10^3^–10^6^	Italy
[48]	fresh cheeses (soft cheese) (v + s)	30.5% (18/59)	<10^3^–10^4^	Italy
[48]	fresh ricotta (v + s)	33.3% (11/33)	<10^3^–10^3^	Italy
[48]	salted ricotta (v + s)	26.9% (14/52)	<10^3^–10^6^	Italy
[48]	seasoned cheeses (v + s)	35% (14/40)	<10^3^–10^6^	Italy
[49]	artisanal mexican cheese (v + s)	29.48% (23/78)	-	Mexico
[50]	buffalo mozzarella (v + s)	26.2% (89/340)	2.2 × 10^2^–3.6 × 10^6^	Italy
[33]	cheese *(Damietta*, *Ras and Kariesh)* (v + s)	95% (71/75)	6.0 × 10^3^–4.2 × 10^6^	Egypt
[51]	dairy products (milk, butter and cheese) (v + s)	6% (5/85)	<10^3^	Morocco

* s = selective detection/count of spores, v + s = count of vegetative cells and spores.

**Table 3 foods-11-02572-t003:** Growth ability of *B. cereus* in different milk and milk products.

Dairy Product	Temperature	Increase (log cfu/g)	Time	Reference
Raw milk	15 °C	<0.5	5 days	[60]
Pasteurized milk	15 °C	3.08–3.11	5 days	[60]
Pasteurized milk	30 °C	>6.0	4 days	[57]
Reconstituted milk	12 °C	>4.0	12.5 days	[61]
Reconstituted milk	22 °C	>4.0	2.5 days	[61]
Reconstituted milk	30 °C	>4.0	1.6 days	[61]
Reconstituted milk	42 °C	>4.0	1.6 days	[61]
Yoghurt	15 °C	0.40–0.90	5 days	[60]
Ricotta	15 °C	5.46–6.52	5 days	[62]
Ricotta	10 °C	2.98–3.91	7 days	[62]
Ricotta salata	4 °C	<0.5	30, 60, 90 days	[9]
Brie	4 °C	<0.5	-	[64]
Brie	8 °C	<0.5	-	[64]
Gouda cheese	30 °C (During production)	2.2	4 h	[65]
Gouda cheese	4–7 °C (During ripening)	<0.5	From 1.6 to 42 days	[65]
Nonfat hard cheese	30 °C	>0.5	30 days	[57]
Mascarpone cheese	15 °C	4.11–4.23	5 days	[60]
Primo sale cheese	15 °C	2.55–4.11	3 days	[56]
Taleggio cheese	15 °C	<0.5	5 days	[60]

## Data Availability

Data is contained within the article.

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
