# Peer review of "Bacillus cereus in Dairy Products and Production Plants"

_foods, 2022, doi:10.3390/foods11172572_

Round 1

Reviewer 1 Report

The manuscript, organized as a Review, is an interesting contribution to dairy microbiology since B. cereus is an organism that has severe technological and health implications. The writing is well organized, with good information, with acceptable English and a good number of references. Some formal issues must be corrected (see below) and some methodological information must be supplied to the speech (also see below):

-line 47: B. cereus strains are able....

-lines 68, 122, 150, 209: "B. cereus" must be distinguished from the rest of the sentence. If the sentence is italicized, the organism name should not be italicized. I suggest not to write the sentences in italics and always keep italics for "B. cereus"

-  lines 100, 121, 207, 231 and 431: B. cereus, in italics

- line 139: references, in chronological order

- lines 167, 181, 213: Table 2, 3... (not "table")

-  line 301: replace "Kumari and Sarkar (2016)" with "These authors applied..."

- line 338: replace "some studies" with "a study"

- In the manuscript, % are permanently mentioned, but it is not clear if they are % of samples. This should be clear

- Another issue that is not clear throughout the manuscript is that when B. cereus count values are given, it is not known whether they are counts of spores or vegetative cells. This must be clarified for a proper interpretation of the count.

- Finally, the review must have a section where the problem of B. cereus counts is discussed, like any other spore-forming bacterium. Since it is very likely that there is a mixture of spores and vegetative cells in the populations, the methodologies to count both must be included (recommended culture media, heat treatment of the sample to then count spores, etc). Are there recommended methods for a selective, differential count...?

Author Response

Reviewer 1

-line 47: B. cereus strains are able....

This was done

-lines 68, 122, 150, 209: "B. cereus" must be distinguished from the rest of the sentence. If the sentence is italicized, the organism name should not be italicized. I suggest not to write the sentences in italics and always keep italics for "B. cereus"

This was done

-  lines 100, 121, 207, 231 and 431: B. cereus, in italics

This was done

- line 139: references, in chronological order

This was done

- lines 167, 181, 213: Table 2, 3... (not "table")

This was done

-  line 301: replace "Kumari and Sarkar (2016)" with "These authors applied..."

This was done

- line 338: replace "some studies" with "a study"

This was done

- In the manuscript, % are permanently mentioned, but it is not clear if they are % of samples. This should be clear

Yes are % on the total of analyzed samples; this was clarified

- Another issue that is not clear throughout the manuscript is that when B. cereus count values are given, it is not known whether they are counts of spores or vegetative cells. This must be clarified for a proper interpretation of the count.

This was included in table 2

- Finally, the review must have a section where the problem of B. cereus counts is discussed, like any other spore-forming bacterium. Since it is very likely that there is a mixture of spores and vegetative cells in the populations, the methodologies to count both must be included (recommended culture media, heat treatment of the sample to then count spores, etc). Are there recommended methods for a selective, differential count...?

This was included

Reviewer 2 Report

The manuscript explores B. cereus pathogenicity. However it is weak as it is, there is a lot of repeated sentences, and English revision is necessary. In addition, the manuscript is lacking of novelty. What is the contribution of the work to the field? The abstract should be concise and only one paragraph. Finally, a review paper must have interesting figures to call attention to the work,  some useful figures could improve paper quality.

Author Response

Reviewer 2

The manuscript explores B. cereus pathogenicity. However it is weak as it is, there is a lot of repeated sentences, and English revision is necessary.

English was revised

In addition, the manuscript is lacking of novelty. What is the contribution of the work to the field?

We think there are pretty much new interesting information included in this review, different from reviews published, especially in table 2 and 3, with prevalences in different products around the world and counts enumerated.

The abstract should be concise and only one paragraph.

This was done

Finally, a review paper must have interesting figures to call attention to the work, some useful figures could improve paper quality.

We included a graphical abstract

Reviewer 3 Report

Manuscript ID: foods-1842411

Entitled: Bacillus cereus in dairy products and production plants

In the current review the authors studied the spore-forming Bacillus cereusis a common contaminant of foods, including dairy products. It is also well known for causing food-borne infections due to the ingestion of food contaminated by live cells, toxins or both, resulting in emetic or diarrheal syndromes. B. cereus is widespread in the environment and can contaminate milk at the time of milking, but it can also reach the dairy products in each phase of production and ripening. Milk pasteurization treatment is not effective in reduce contamination and can instead act as an activator of their germination; thus, a potential risk associated still exists with the consumption of some processed foods.

After reviewing the current manuscript I found that this manuscript can be accepted after minor revision 

1- the abstract should be modified to be clear for the reader 

2- the English should be modified to eiminate any errors 

3- the conclusion part should included the future outlook   

4- All Tables must have the caption over the Table not under it (Table 1, Table2 and Table 3)  

Author Response

Reviewer 3

1- the abstract should be modified to be clear for the reader 

This was done

2- the English should be modified to eiminate any errors 

This was done

3- the conclusion part should included the future outlook   

This was done

4- All Tables must have the caption over the Table not under it (Table 1, Table2 and Table 3)  

This was done

Reviewer 4 Report

Overview

The review focuses on the implications of the spore-forming bacterium Bacillus cereus as a contaminant of milk and dairy products. This microorganism has both a technological interest, as it can cause the spoilage of these foods, and a healthy concern as an agent of food-borne infections due to the ingestion of food contaminated by live cells, toxins or both. Bacillus cereus is widespread in the environment and can contaminate milk and dairy products in each phase of production, storing and ripening. Milk pasteurisation can act as an activator of spore germination. Prevalences and concentrations of B. cereus in milk and dairy products are also reviewed. Another issue that is addressed is the production of biofilms by this microorgansm, what represents a serious problem for dairy industry. It is concluded that correct application of hygienic procedures is extremely important to avoid and control the circulation of B. cereus along the dairy supply chain, and that future interventions must consider the synergic application of different mild technologies (hurdle technology) to prevent biofilm formation and to remove the microorganism on the dairy equipment.

In my opinion, although the review is straightforward and generally well written, there are numerous formal deficiencies that should be corrected before publishing the article in Foods journal.

Comments are provided to the authors.

General comments

- The English language should have been revised by a native English speaker with some knowledge of the subject. Either UK or US English should be used, but the two styles should never be mixed.

- Spaces between words, before units or after marks should be checked.

- Avoid starting a sentence with an abbreviated word such as the initial of a genus name.

- The meaning of the abbreviations must always be specified and mentioned in its entirety the first time the acronyms are used.

- Unit abbreviations (cfu, mL) should be spelled correctly.

Detailed comments

Abstract

- p. 1. Lines 14, 23 and 26, and throughout the text. Avoid starting a sentence with an abbreviated word (genus name). Change to “Bacillus cereus”.

- p. 1. Line 16. The storage stage should also be included: “production, storage and ripening”.

- p. 1. Line 16. Change to “pasteurisation” if using UK spelling.

- p. 1. Line 17. Specify that the pasteurisation treatment can act as an “activator of spore germination”.

- p. 1. Lines 20 and 21. Change the letter “x” to the symbol “×”, here and throughout the text.

Section 1

- p. 1. Line 41. Avoid starting a sentence with the abbreviated genus name. Change to “Bacillus cereus”, here and throughout the text.

- p. 1 Line 44. Change the font size and style in “2018” and “et al., 2019”.

- p. 2. Line 47. Change “is” to “are”.

- p. 2. Line 56. Describe the “FSANZ” abbreviation. Change to e.g. “Food Standard Australia New Zealand, FSANZ, 2013”.

- p. 2. Line 63. Better change the order to “Extracellular Polymeric Substances, EPS”.

- p. 2. Line 77. Change “germs” to e.g. “cells” or “bacteria”.

- p. 2. Line 80. Change to “Haemolysin” if you are using British spelling. Be consistent.

- p. 2. Lines 82–83. Change the order of the references (alphabetical order).

- p. 2. Line 97. Change to “…vary from 103 to 105 ufc/g (European Food Safety Authority, EFSA, 2016).”

- p. 2. Line 99. Change to “Granum & Baird Parker, 2000”.

- p. 3. Table 1. Change to “HBL, NHE, CytK” as in the text. Change to “ufc” (lowercase letters) as in the text. Be consistent.

- p. 3. Line 108. Change to “log” (lowercase initial), here and throughout the text.

- p. 3. Lines 114–115. Change to “…can also be used as ingredients (vegetables, spices) in the food industry…”

Section 2

- p. 3. Line 121. Write “B. cereus” in italics. Write “dairy” with lowercase initial.

- p. 4. Line 131. Change “products” to “product”.

- p. 4. Line 141. Change to “Svensson et al., 2001, 2004”.

- p. 4. Line 160. Change to “e.g.”

- p. 4. Line 171. Change to e.g. “25.0 to 62.8%”. Always use the same number of decimal places and the same number style in comparisons.

- p. 4. Lines 172–173. Ice cream manufacturing operations usually include pasteurization or heating of the mixture of ingredients.

- p. 5. Line 183. Change to “log” (lowercase initial), here and throughout the text.

- p. 5. Line 187. Change to e.g. “European Commission, EC, 2005”.

- p. 5. Line 195. Change to “most strains isolated from dairy products …”

- p. 5. Line 206. Write “nhe” in italics.

- p. 5. Table 2. Change to “Pasteurised milk” (and “pasteurisation”), here and throughout the text, if using UK spelling. Be consistent.

- p. 5. Table 2. Change to “Raw milk” and “Pasteurised milk” (lowercase initial of “milk”). Change to “te Giffel” (lower case initial of “te”).

- p. 5 Table 2. “Loads (cfu/g)” column. Always use the same number of decimal places and the same number style in comparisons. Change the letter “x” to the symbol “×”, here and throughout the text. As examples, change to “4.0×101-3.8×105” or “1.0×101-2.2×102”.

- p. 6 Table 2. Change to “Pasteurised buffalo milk”. Change to “Fermented milk (nunu)” (it is not a yogurt).

- p. 6 Table 2. “Loads (cfu/g)” column. Always use the same number of decimal places and the same number style in comparisons. Change the letter “x” to the symbol “×”, here and throughout the text. As an example, change to “1×101-5×103”. Check for “5×100”.

- p. 7. Table 2. Write “Damietta”, “Ras” and “Kariesh” with the initials capitalized.

- p. 7. Table 2. “Loads (cfu/g)” column. Always use the same number of decimal places and the same number style in comparisons. Change the letter “x” to the symbol “×”.

- p. 7. Table 2. Write “B. cereus” in italics in the title of the table.

- p. 8. Lines 216–218. Write “lactic acid bacteria” with lowercase initials, also on line 226. Could milk pasteurization destroy or inactivate natural bacterial inhibitors, such as agglutinins or lactoperoxidase?

- p. 8. Line 227. Change to “physical-chemical conditions” or “physicochemical conditions”. Change to “pH 5.8-6.8” and “pH 5.30-6.32”.

- p. 8. Table 3. Change to “Nonfat hard cheese” (lowercase initials). Change “curing” to “ripening”. Write “B. cereus” in italics in the title of the Table.

- p. 9. Line 242. Change to “source of spore contamination”.

- p. 9. Line 244. Change to “after 48 h incubation”.

- p. 9. Line 245. Change “meaning” to e.g. “indicating”.

- p. 9. Line 254. Change to “in the presence of antagonistic substances”.

- p. 9. Line 260. Change “fifty-six” to “56”.

- p. 9. Line 271. Change “undomesticated” to e.g. “wild”.

- p. 9. Line 280. Change to “stainless-steel tube”.

- p. 10. Lines 303 and 305. Join the degree symbol (º) with the number, as in the rest of the text. Be consistent.

Section 3

- p. 10. Line 317. Change to “log cfu/mL”.

- p. 11. Line 339. Does a reduction “of about 100 times” mean a reduction “of 2 log units”?

- p. 11. Line 346. “primo sale” or “Primo Sale” cheese?

- p. 11. Line 354. Change to “sterilisation” if using UK spelling.

Section 4 (Conclusions)

- p. 11. Line 376. Change to “Good Manufacturing Practices” (capital initials).

References

- pp. 12–15. Thoroughly check that the references conform to the formatting style specified in the instructions for the authors of the journal.

Author Response

 The English language should have been revised by a native English speaker with some knowledge of the subject. Either UK or US English should be used, but the two styles should never be mixed.

This was done

-Spaces between words, before units or after marks should be checked.

This was done

-Avoid starting a sentence with an abbreviated word such as the initial of a genus name.

This was done

- The meaning of the abbreviations must always be specified and mentioned in its entirety the first time the acronyms are used.

This was done

- Unit abbreviations (cfu, mL) should be spelled correctly.

This was done

Detailed comments

Abstract

- p. 1. Lines 14, 23 and 26, and throughout the text. Avoid starting a sentence with an abbreviated word (genus name). Change to “Bacillus cereus”.

This was done

- p. 1. Line 16. The storage stage should also be included: “production, storage and ripening”.

This was done

- p. 1. Line 16. Change to “pasteurisation” if using UK spelling.

This was done

- p. 1. Line 17. Specify that the pasteurisation treatment can act as an “activator of spore germination”.

This was done

- p. 1. Lines 20 and 21. Change the letter “x” to the symbol “×”, here and throughout the text.

This was done

Section 1

- p. 1. Line 41. Avoid starting a sentence with the abbreviated genus name. Change to “Bacillus cereus”, here and throughout the text.

This was done

- p. 1 Line 44. Change the font size and style in “2018” and “et al., 2019”.

This was done

- p. 2. Line 47. Change “is” to “are”.

This was done

- p. 2. Line 56. Describe the “FSANZ” abbreviation. Change to e.g. “Food Standard Australia New Zealand, FSANZ, 2013”.

This was done

- p. 2. Line 63. Better change the order to “Extracellular Polymeric Substances, EPS”.

This was done

- p. 2. Line 77. Change “germs” to e.g. “cells” or “bacteria”.

This was done

- p. 2. Line 80. Change to “Haemolysin” if you are using British spelling. Be consistent.

This was done

- p. 2. Lines 82–83. Change the order of the references (alphabetical order).

This was done

- p. 2. Line 97. Change to “…vary from 103 to 105 ufc/g (European Food Safety Authority, EFSA, 2016).”

This was done

- p. 2. Line 99. Change to “Granum & Baird Parker, 2000”.

This was done

- p. 3. Table 1. Change to “HBL, NHE, CytK” as in the text. Change to “ufc” (lowercase letters) as in the text. Be consistent.

This was done

- p. 3. Line 108. Change to “log” (lowercase initial), here and throughout the text.

This was done

- p. 3. Lines 114–115. Change to “…can also be used as ingredients (vegetables, spices) in the food industry…”

This was done

Section 2

- p. 3. Line 121. Write “B. cereus” in italics. Write “dairy” with lowercase initial.

This was done

- p. 4. Line 131. Change “products” to “product”.

This was done

- p. 4. Line 141. Change to “Svensson et al., 2001, 2004”.

This was done

- p. 4. Line 160. Change to “e.g.”

This was done

- p. 4. Line 171. Change to e.g. “25.0 to 62.8%”. Always use the same number of decimal places and the same number style in comparisons.

This was done

- p. 4. Lines 172–173. Ice cream manufacturing operations usually include pasteurization or heating of the mixture of ingredients.

This was rephrased

- p. 5. Line 183. Change to “log” (lowercase initial), here and throughout the text.

This was done

- p. 5. Line 187. Change to e.g. “European Commission, EC, 2005”.

This was done

- p. 5. Line 195. Change to “most strains isolated from dairy products …”

This was done

- p. 5. Line 206. Write “nhe” in italics.

This was done

- p. 5. Table 2. Change to “Pasteurised milk” (and “pasteurisation”), here and throughout the text, if using UK spelling. Be consistent.

This was done

- p. 5. Table 2. Change to “Raw milk” and “Pasteurised milk” (lowercase initial of “milk”). Change to “te Giffel” (lower case initial of “te”).

This was done

- p. 5 Table 2. “Loads (cfu/g)” column. Always use the same number of decimal places and the same number style in comparisons. Change the letter “x” to the symbol “×”, here and throughout the text. As examples, change to “4.0×101-3.8×105” or “1.0×101-2.2×102”.

This was done

- p. 6 Table 2. Change to “Pasteurised buffalo milk”. Change to “Fermented milk (nunu)” (it is not a yogurt).

This was done

- p. 6 Table 2. “Loads (cfu/g)” column. Always use the same number of decimal places and the same number style in comparisons. Change the letter “x” to the symbol “×”, here and throughout the text. As an example, change to “1×101-5×103”. Check for “5×100”.

This was done

- p. 7. Table 2. Write “Damietta”, “Ras” and “Kariesh” with the initials capitalized.

This was done

- p. 7. Table 2. “Loads (cfu/g)” column. Always use the same number of decimal places and the same number style in comparisons. Change the letter “x” to the symbol “×”.

This was done

- p. 7. Table 2. Write “B. cereus” in italics in the title of the table.

This was done

- p. 8. Lines 216–218. Write “lactic acid bacteria” with lowercase initials, also on line 226. Could milk pasteurization destroy or inactivate natural bacterial inhibitors, such as agglutinins or lactoperoxidase? yes, this was included

This was done

- p. 8. Line 227. Change to “physical-chemical conditions” or “physicochemical conditions”. Change to “pH 5.8-6.8” and “pH 5.30-6.32”.

This was done

- p. 8. Table 3. Change to “Nonfat hard cheese” (lowercase initials). Change “curing” to “ripening”. Write “B. cereus” in italics in the title of the Table.

This was done

- p. 9. Line 242. Change to “source of spore contamination”.

This was done

- p. 9. Line 244. Change to “after 48 h incubation”.

This was done

- p. 9. Line 245. Change “meaning” to e.g. “indicating”.

This was done

- p. 9. Line 254. Change to “in the presence of antagonistic substances”.

This was done

- p. 9. Line 260. Change “fifty-six” to “56”.

This was done

- p. 9. Line 271. Change “undomesticated” to e.g. “wild”.

This was done

- p. 9. Line 280. Change to “stainless-steel tube”.

This was done

- p. 10. Lines 303 and 305. Join the degree symbol (º) with the number, as in the rest of the text. Be consistent.

This was done

Section 3

- p. 10. Line 317. Change to “log cfu/mL”.

This was done

- p. 11. Line 339. Does a reduction “of about 100 times” mean a reduction “of 2 log units”? yes

- p. 11. Line 346. “primo sale” or “Primo Sale” cheese?

“Primo Sale” cheese

- p. 11. Line 354. Change to “sterilisation” if using UK spelling.

This was done

Section 4 (Conclusions)

- p. 11. Line 376. Change to “Good Manufacturing Practices” (capital initials).

This was done

References

- pp. 12–15. Thoroughly check that the references conform to the formatting style specified in the instructions for the authors of the journal.

This was done

Round 2

Reviewer 1 Report

In my opinion, the authors have responded satisfactorily to the obnservations made

Reviewer 2 Report

The authors made all the corrections requested